# Epidemiological and Molecular Surveillance of Aichi Virus A at Different Stages of Sewage Treatment: A One-Year Study in the Southeast of Brazil

**DOI:** 10.3390/v17050736

**Published:** 2025-05-21

**Authors:** Mariah C. A. do Nascimento, Camila R. Rosa, Meriane Demoliner, Dayla B. Geraldini, Guilherme R. F. Campos, Daniela M. Quevedo, Rafael N. Miceli, Fernando R. Spilki, João Pessoa Araújo, Marilia F. Calmon, Paula Rahal

**Affiliations:** 1Department of Biology, São Paulo State University (UNESP), São José do Rio Preto 15054-000, SP, Brazil; mariahantunees@gmail.com (M.C.A.d.N.); camila.r.rosa@unesp.br (C.R.R.); daylageraldini@gmail.com (D.B.G.); p.rahal@unesp.br (P.R.); 2Molecular Microbiology Laboratory, University Feevale, Novo Hamburgo 93525-075, RS, Brazil; merianedemoliner@gmail.com (M.D.); fernandors@feevale.br (F.R.S.); 3Department of Dermatological, Infectious and Parasitic Diseases, Medical School of São José do Rio Preto (FAMERP), São José do Rio Preto 15090-000, SP, Brazil; guilhermecampos07@gmail.com; 4Institute of Exact and Technological Sciences (ICET), University Feevale, Novo Hamburgo 93525-075, RS, Brazil; danielamq@feevale.br; 5SeMAE—Autonomous Municipal Water and Sewage Service, São José do Rio Preto 15048-000, SP, Brazil; rnmiceli@gmail.com; 6Biotechnology Institute, São Paulo State University (UNESP), Botucatu 18618-687, SP, Brazil; joao.pessoa@unesp.br

**Keywords:** enteric viruses, sewage treatment, infectivity assay

## Abstract

Enteric viruses, such as the Aichi virus (AiV), pose a potential health risk due to their high excretion rates through fecal elimination, limited removal during treatment processes, and prolonged survival, highlighting the need to assess the potential for exposure and disease transmission through sanitation systems. This study investigated the prevalence of AiV at three key stages of sewage treatment in the city of São José do Rio Preto, São Paulo state, Brazil, as well as its viral concentrations, infectious potential, and molecular characterization. The data were also analyzed for potential correlations with reported diarrheal disease cases in the city and the physicochemical properties of sewage. The methodology employed included Nested PCR, qPCR, Sanger Sequencing, and phylogenetic analysis, as well as infectivity testing in cell cultures. The prevalence of AiV throughout the year in raw sewage samples was 90.4%, 78.8% in post-anaerobic biological treatment, and 71.1% in post-chemical treatment, totaling 125 positive samples out of 156, being characterized as AiV genotype A. The virus also demonstrated persistence and infectious potential at all three stages analyzed. The AiV-A mean concentration ranged from 2.05 log^10^ to 4.64 GC/mL, 2.31 to 4.72 log^10^ GC/mL, and 2.13 to 2.85 log^10^ GC/mL for the same treatment stages, respectively. A significant difference (*p* ≤ 0.05) suggests higher viral concentrations in summer at the three sewage process points analyzed, while lower viral concentrations were observed in post-chemical treatment samples (*p* ≤ 0.01). Additionally, no statistically significant relationship was observed between the virus occurrence in samples and cases of acute diarrheal diseases in the city. In conclusion, this study highlights that much remains to be understood about AiV while providing valuable insights into the relationship between AiV, environmental factors, and public health.

## 1. Introduction

Sewage consists of various water sources, including municipal, industrial, infiltration, and stormwater, forming a complex matrix derived from urine, blood, vomit, sputum, and feces [1]. These waters are directed to sewage treatment plants (STPs), which are facilities responsible for removing most raw chemical and biological pollutants through mechanical, biological, and chemical processes [2]. The mechanical process entails separating solid elements such as debris and sand, while biological treatment employs microorganisms that digest biodegradable organics into simpler, more stable substances, such as gases and biomass. Lastly, the chemical process involves disinfecting the effluent with chlorine [3,4].

Enteric viruses, emerging pathogens responsible for disease outbreaks globally, present a potential health risk even at low levels of exposure [5,6]. These viruses are excreted by infected individuals, discharging 10^2^ to 10^5^ viral particles per gram of fecal material into the sewage treatment system, posing greater health risks compared to pathogenic bacteria or protozoa [7,8]. Due to their high excretion, limited removal in treatment processes, and prolonged survival, these viruses are significantly present in the environment, and given their significant health threats and potentially serious consequences, it is necessary to assess the potential for exposure and transmission of diseases through sanitation systems [8,9]. The main etiological agents of acute non-bacterial gastroenteritis worldwide include human norovirus (NoV), rotavirus (RoV), human adenovirus (HAdV), hepatitis A virus (HAV), enteroviruses (EV), and astrovirus (AsV) [6,7,8,10]. Other viruses like human Aichi virus (AiV), first detected in Japan in 1989, have also emerged as causes of gastroenteritis outbreaks [11].

AiV, a non-enveloped, single-stranded RNA (ssRNA) virus from the *Picornaviridae* family, exhibits icosahedral symmetry. The virions are around 30 nm in diameter, and the species *Kobovirus aichi* (also known as *Aichivirus A*) is classified into three genotypes (AiV-A, AiV-B, and AiV-C) within the human population [11,12]. Some clinical signs caused by AiV infection include diarrhea, abdominal pain, nausea, vomiting, and fever. However, its pathogenesis is more pronounced in subclinical infections. The virus also exhibits an opportunistic nature, being highly prevalent in immunodeficient individuals [11,13]. It can contaminate aquatic environments directly or via raw or treated sewage discharge, acting as a causative agent of gastroenteritis and is primarily transmitted through the fecal–oral route [14]. In Brazil, the first detection in human children’s feces was reported in 2006 [15]. However, this virus has been globally detected in diverse environments such as sewage [16,17,18,19,20,21,22], surface waters [17,23], rivers [16,20], and beach sand [24]. Additionally, AiV demonstrates resistance to various treatment methods and can persist in the environment for extended periods, emphasizing the critical importance of environmental surveillance for monitoring it [14,25].

The use of wastewater to understand community health and exposure is not a new concept. Wastewater-Based Epidemiology (WBE) has proven to be a valuable tool in complementing public health strategies by filling epidemiological gaps and providing crucial information on the current prevalence of infections in the human population. In some cases, it also serves as an early warning of viral spread within communities [26,27,28]. Such approaches are particularly relevant in areas with high-quality sanitation systems, such as São José do Rio Preto, São Paulo state, where this present study was conducted. According to the Trata Brazil Institute, the city was ranked first in sanitation in 2023 [29] and second in 2024 [30], making it an ideal location to apply WBE in monitoring public health. Thus, the objective of this study is to examine the prevalence, viral concentration, and infectivity potential of AiV and molecularly characterize it in positive samples obtained at three distinct sewage treatment points in São José do Rio Preto, São Paulo state, Brazil. Additionally, this study aims to correlate these findings with the epidemiological data of the city and the physicochemical properties of sewage.

## 2. Materials and Methods

### 2.1. Characterization of the Study Area and Sampling

Located in the interior of the state of São Paulo, Brazil, the municipality of São José do Rio Preto covers an area of 431.944 km^2^ and has approximately 480,393 inhabitants (Figure 1), according to the latest updated data from the Brazilian Institute of Geography and Statistics (IBGE) in 2022 [31]. In São José do Rio Preto, São Paulo state, temperatures typically range between 20.8 and 27.4 °C. Autumn and winter bring cooler and drier conditions, whereas spring and summer are marked by warmer temperatures and higher precipitation, as detailed in Appendix A [32].

Samples for the study were collected in partnership with the São José do Rio Preto Sewage Treatment Plant, overseen by the Municipal Water and Sewage Service (SeMAE), which handles 100% of the sewage of the municipality. Twenty-four-hour 500 mL composite samples were collected at three key points within the treatment process: raw sewage (RS), post-anaerobic biological treatment (PABT), and post-chemical treatment (PCT) with chlorine, as depicted in Appendix A [4]. Sampling was conducted weekly for one year (52 weeks), from 29 March 2022 to 20 March 2023. The HACH Sigma SD900 AWRS refrigerated automatic sampler (HACH, Loveland, CO, USA) was used, with the sampling program adjusted to the input flow. A total of 156 samples were collected. The samples gathered in this study were labeled according to the detection stage, with the week of collection (1–52) following the stage. Once collected, the samples were kept at below 4 °C and sent to São Paulo State University Laboratory in glass flasks on ice. Upon receipt, for the initial clarification procedure, 222 mL of each sample underwent centrifugation at 3000× *g* at 4 °C for 20 min to remove suspended material. Subsequently, following the protocol by Girardi et al. (2018) [33], with some modifications, samples were concentrated using ultracentrifugation (Beckman Coulter, Indianapolis, IN, USA) at 41,000× *g* at 4 °C for 3 h. The resulting pellets were resuspended in Tris-EDTA buffer (pH 8.0) to a final volume of 2 mL and homogenized. Samples were aliquoted and stored at −8 °C until extraction.

### 2.2. RNA Extraction and cDNA Synthesis

The RNA from 250 μL of the final concentrate of each sample was extracted using TRIzol reagent (Thermo Fisher Scientific, Waltham, MA, USA) as per the manufacturer’s instructions. After elution in 30 µL of DEPC-treated water (Sigma-Aldrich, Saint Louis, MI, USA), the RNA was stored at −80 °C. For cDNA synthesis, a High-Capacity cDNA Synthesis Kit (Applied Biosystems, Waltham, MA, USA) was employed following the manufacturer’s protocol, and the resulting cDNA was stored at −20 °C.

### 2.3. Construction of the Control Plasmid

For polymerase chain reaction (PCR) and Nested PCR positive control, a plasmid was obtained from Integrated DNA Technologies (IDT; Coralville, IA, USA), containing inserts of sequences from regions targeted by the PCR primers of AiV, as described in Appendix A [15,22,34]. The reference sequence was taken from NCBI (Accession number: NC_001918). The plasmid was purified using the GeneJET Plasmid Miniprep Kit (Thermo Fisher Scientific, USA), and the concentration of purified plasmids was determined by measuring the optical density at 260 nm using a NanoDrop 2000 spectrophotometer (Thermo Fisher Scientific, USA).

### 2.4. Nested PCR

Nested PCR reactions were carried out using the GoTaq^®^ Colorless Master Mix kit (Promega, Madison, WI, USA). Each reaction mixture, totaling 25 µL, consisted of 12.5 µL of 2 × Master Mix, 400 nM of each forward and reverse primer, 9 µL of nuclease-free water, and 2.5 µL of nucleic acid for PCR and the PCR product for Nested PCR. The primer sequence and cycling parameters are shown in Appendix A [15,22,34]. The plasmid DNA served as the positive control, whereas for the no-template control (NTC), ultrapure water was added instead of nucleic acid. Amplification was performed using a Veriti 96-well thermocycler (Applied Biosystems, USA). After amplification, PCR products were analyzed on a 2% agarose gel containing 0.5 µg/mL ethidium bromide. Electrophoresis was conducted at 90 V for 40 min. The molecular sizes of the products were compared against a 100 bp DNA ladder (Thermo Fisher Scientific, USA). Bands were visualized under ultraviolet light, and images were captured using the L-Pix Touch photo documentation system (Loccus, Cotia, São Paulo, Brazil).

### 2.5. Sanger Sequencing and Phylogenetic Analysis

Nested PCR products from AiV-positive samples were purified using magnetic beads (SpeedBead Magnetic Carboxylate Modified Particles, Cytiva, Sigma-Aldrich, USA), following the manufacturer’s instructions. For sequencing, the BigDye Terminator v3.1 methodology (Applied Biosystems, USA) was employed using both Nested PCR primers, listed in Appendix A [15,22,34]. Sequencing was performed using the Sanger method on the automated sequencer Spectrum Compact CE System (Promega, USA).

The contigs were assembled by aligning pairs of forward and reverse Sanger sequences using Geneious Prime^®^ version 2023.2.1. To perform the phylogenetic analysis, 3C region sequences of AiV were selected from GenBank. Alignments were generated in Geneious Prime^®^ using the MUSCLE algorithm [35]. The phylogenetic tree was built using the IQ-TREE version 1.6.12 [36] applying the Maximum Likelihood method and the TIM3 + F + G4 substitution model, which was identified as the best-fit model, with 1000 bootstrap replicates to assess tree reliability. The resulting tree was further refined in MEGA version 10.1.7. Sequences exhibiting a high number of degenerate positions or poor quality were excluded from the analysis.

### 2.6. Infectivity Assay

An infectivity assay was carried out to analyze the infectious potential of AiV.

Two concentrated samples from each treatment stage (i.e., RS-21 and RS-43, PABT-2 and PABT-46, and PCT-10 and PCT-13), totaling 1 mL per treatment stage, underwent ultracentrifugation once more at 41,000× *g* for 3 h (Beckman Coulter) at 4 °C. Following centrifugation, the supernatant was removed, and the pellet was resuspended in 2 mL of DMEM culture medium supplemented with 2% fetal bovine serum. Subsequently, the samples were filtered using 0.45 µM and 0.22 µM filters, respectively, in a Biosafety Level 3 laboratory (BSL-3). The samples from the three different treatment stages were selected based on their positive results for AiV in Nested PCR and on sample availability.

For viral isolation, the day before, VERO cells (ATCC CCL-81) were cultured in 25 cm^2^ flasks with DMEM medium supplemented with 10% fetal bovine serum, 100 U/mL penicillin, and 100 µg/mL streptomycin to achieve 90% to 95% confluence on the day of infection. In the BSL-3 lab, the culture medium was removed, and 300 µL of each filtered sample, totaling three flasks (one for each treatment stage), was added to the cell monolayer. The cells were incubated for 1 h at 37 °C with periodic shaking of the flasks every 5 min to ensure maximum cell–virus contact and prevent drying of the cell monolayer. Following incubation, 5 mL of DMEM medium containing 2% fetal bovine serum, 100 U/mL penicillin, and 100 µg/mL streptomycin was added to each flask, followed by further incubation for 120 h or until cytopathic effects appeared. After incubation or the onset of cytopathic effects, infected cells were frozen at −80 °C to lyse all infected cells, thereby optimizing viral isolation by releasing viral particles still present inside the cells. After freezing, the supernatant was thawed, centrifuged at 1000 rpm for 5 min to precipitate cellular debris, aliquoted, and stored at −80 °C. Three successive passages were conducted for each sample, and confirmation of viral isolation was achieved by detecting viral cDNA in the supernatant during each passage using Nested PCR and sequencing, as illustrated in Figure 2.

### 2.7. Quantitative PCR (qPCR)

Quantification of AiV cDNA was performed using qPCR, specifically for genotype A (AiV-A) of the species *Kobovirus aichi* (*Aichivirus A*), based on the results of the phylogenetic analysis. A standard curve was established by preparing ten-fold serial dilutions of gBlock (IDT, USA), covering the region of the F and R primers for qPCR, as outlined in Appendix A [15,22,34], ranging from 1 × 10^1^ to 1 × 10^6^ genomic copies (GC)/µL. Reactions were carried out using the QuantiNova Probe PCR Master Mix (Qiagen, Hilden, Germany). Each 10 µL reaction contained 5.0 µL of 2 × Master Mix, 1:200 dilution of Dye Rox, 800 nM of each forward and reverse primer, 300 nM of the probe, 2.05 µL of nuclease-free water, and 1.0 µL of nucleic acid. Additional information on the primers, probe, and cycling conditions can be found in Appendix A [15,22,34]. The qPCR was performed on a QuantStudio 12K Flex instrument (Applied Biosystems, USA), with manual settings for threshold and baseline. All reactions were conducted in triplicate.

### 2.8. qPCR Inhibition

A PCR inhibition assay was performed on all 156 cDNA samples derived from wastewater using the Sketa22 real-time PCR assay [35]. Each sample received a specific amount (10^4^/reaction) of *Oncorhynchus keta* (*O. keta*) gBlock DNA (IDT, USA) to assess potential inhibition. To establish a baseline, *O. keta* DNA was also added to DNase- and RNase-free water, and the average Cq value was recorded. PCR inhibition was considered present if the cycle threshold (Ct) increased by more than two cycles after the addition of nucleic acids, following the criteria outlined by Staley et al. (2012) [37].

### 2.9. Data Analysis

Descriptive analyses, including percentages, minimum and maximum values, measures of central tendency (mean and median), and measures of dispersion (standard deviation), were represented graphically using Microsoft Excel version 2021 (Microsoft Corporation, Redmond, WA, USA) or SPSS version24 (IBM, Armonk, NY, USA).

The Kolmogorov–Smirnov test was applied to assess normality, and since the data did not follow a normal distribution, non-parametric tests were used. For normalization, a reference flow for AiV was followed as described in the study by Nagarkar et al. (2022) [38]. To identify the correlation with acute diarrheal disease (ADD) in São José do Rio Preto, São Paulo state, and the hydrological data, instantaneous correlation (lag = 0) was evaluated in the RS samples. The hydrological data related to the samples were provided by SeMAE, while the ADD cases were accessed on the Ministry of Health’s website [39]. The cross-correlation function also assessed the AiV correlation across various lags, with each lag representing a week, consistent with the weekly sampling schedule. The chi-square test was applied to examine the relationship between the seasons of the year, sewage parameters, and the concentration of AiV. The Kruskal–Wallis test was then used to identify the parameters or seasons that showed a significant difference between the three stages of sewage treatment, with the Bonferroni correction also applied to account for multiple comparisons. To compare the detection of AiV between rainy and non-rainy days, the Mann–Whitney test was used. The results were considered significant with *p* ≤ 0.05, and statistical analysis was performed using SPSS version24.

Samples were classified as qPCR-positive if they tested positive in at least two of the three replicates. They were classified as qPCR-positive but non-quantifiable (NQ) if the quantification value was below 1 GC/reaction. Additionally, just for statistical analysis, for the non-quantifiable samples, the value was substituted with half of the smallest value found [40].

## 3. Results

### 3.1. Detection of AiV at Different Stages of Sewage Treatment During the Seasons of the Year and Acute Diarrheal Disease (ADD) Reported Cases in the City

Out of the 156 samples analyzed, 125 samples (80.1%) tested positive for AiV in the Nested PCR; 47 (37.6%) were from RS, 41 (32.8%) were from PABT, and 37 (29.6%) were from PCT. Also, considering each phase independently, of the 52 samples collected throughout the year, the detection rates were 90.4%, 78.8%, and 71.1% for the same sewage treatment stages, respectively.

As illustrated in Figure 3, in autumn, AiV was positive in 78.6% of RS samples, 71.4% in PABT, and 57.1% in PCT. In this season, 6986 cases of acute diarrheal diseases (ADD) were reported. In winter, 100% of the samples were positive for AiV at all three treatment stages, while 5744 cases of ADD were reported. In spring, 92.3% of the samples were positive for AiV in RS, 38.5% in PABT, and 38.4% in PCT. A total of 5512 cases of ADD were reported during this period. In summer, 91.7% were positive for AiV in RS and PCT, while 100% were positive in PABT. In this season, 7258 cases of ADD were reported. A total of 25,500 ADD cases were reported in the city of São José do Rio Preto during the studied year.

### 3.2. Sequencing and Phylogenetic Analysis

A total of 125 AiV-positive samples, identified through Nested PCR, were sequenced using Sanger methodology; however, some sequences were of insufficient quality and were excluded from the analysis. A phylogenetic tree was constructed using sequences from both Sanger sequencing and those available in GenBank.

In the phylogenetic analysis, 88 samples were included, using a 124–180 bp consensus fragment. More details about the sequences used are available in Appendix A. Of the eighty-eight samples analyzed phylogenetically, eighty-six (97.7%) were identified as human AiV-A, while two (2.3%) were identified as *Canine kobovirus*, also belonging to the AiV genotype A, as illustrated in Figure 4.

The similarity between the human samples in this study and the reference genome for *Kobovirus aichi* (EF079154.1) ranged from 88.8% to 97.6% (PV101084 (PABT-18) and PV101121 (PCT-34), respectively). For the canine samples, similarity to the reference genome for *Canine kobuvirus* (MH747478.1) was 90.1% and 91.1% for PV101119 (PCT-32) and PV101069 (RS-39), respectively). The lower similarity observed in some samples is likely due to the presence of degenerate bases. Due to the file size, additional information regarding the sample similarity is available online “https://docs.google.com/spreadsheets/d/13wxMhMu4M8Hrh7HOHf5ML1dzD5uZ4qt0/edit?usp=drive_link&ouid=103923547454008704058&rtpof=true&sd=true” (accessed on 2 May 2025).

### 3.3. Performance and Inhibition Test of qPCR Assay

The AiV-A qPCR assay demonstrated an efficiency of 99.51% in its standard curve, with a linearity (R^2^) of 0.996. The Y-intercept was 35.282, and the slope was −3.334. For the inhibition test, out of the 156 samples analyzed, seven (4.5%) showed signs of inhibition. The cDNA from these samples was then diluted 1:10 with DNase- and RNase-free water and retested, with no further inhibition detected.

### 3.4. Quantification of AiV-A at Different Stages of Sewage Treatment

#### 3.4.1. Correlation with ADD Cases and Seasonal Variation

The data were grouped by week and season, using the Southern Hemisphere’s seasonal cycle as the reference, and included the quantification for each sample along with ADD cases, as shown in Figure 5.

High levels of ADD cases were reported throughout the year, with an average of 452.4 ± 50.3 cases from weeks 1–44, except for week 10 (May 2022), which showed a peak, reporting 680 cases. In weeks 45–47 (January and February, 2023), an increase began to be observed, with an average of 579.7 ± 14.3 cases. The following weeks (48–52; February and March, 2023) showed the highest average number of ADD cases observed in this study, with 734.6 ± 90.9 cases.

Out of the 125 samples that tested positive using Nested PCR, 91 were positive by qPCR, of which 77 were quantifiable. AiV-A was detected throughout the seasons in RS samples, with concentrations ranging from 2.05 to 4.64 log^10^ GC/mL. The highest quantification (i.e., RS-48) occurred in summer, while the lowest was observed in autumn (i.e., RS-13). However, five samples were positive but not quantifiable, one (i.e., RS-15) in winter and four (RS-27, RS-35, RS-37, and RS-39) in spring. In the PABT samples, concentrations ranged from 2.31 to 4.72 log^10^ GC/mL, with the highest concentration detected in sample PABT-49 (i.e., summer), and the lowest in samples PABT-12 and PABT-40 (i.e., autumn and summer, respectively). Six samples were not quantifiable at this stage, one (i.e., PABT-13) from autumn, four (i.e., PABT-18, PABT-19, PABT-24, and PABT-25) from winter, and one (i.e., PABT-39) from spring. For PCT samples, concentrations ranged from 2.13 to 2.85 log^10^ GC/mL in samples collected during the summer, with the highest concentration identified in sample PCT-48 and the lowest in PCT-47. However, three samples at this stage were also not quantifiable, one (i.e., PCT-11) from autumn, one (i.e., PCT-26) from winter, and one (i.e., PCT-46) from summer. Additional details on the positive and quantified samples can be found in Appendix A.

Descriptive analyses of AiV-A quantification across seasons are reported in Appendix A. For non-parametric tests, percentile values may be used to assess the data distribution or rankings.

The Chi-square test, used to assess the association between the variables, indicated significance (*p* ≤ 0.01) for the seasons and AiV-A in RS (*χ*^2^ = 12.37; *p* = 0.006), PABT (*χ*^2^ = 22.45; *p* = 0.000), and PCT (*χ*^2^ = 11.84; *p* = 0.008). Therefore, the Kruskal–Wallis test was used for ranking and analyzing these variables at different stages of sewage treatment, as depicted in Figure 6.

When comparing AiV-A quantification across different seasons and all three sewage treatment stages, the Kruskal–Wallis test with a Bonferroni correction, used to prevent false positives, revealed significant differences in RS for spring and summer, with summer showing higher quantifications; in PABT for autumn and spring, with spring showing higher concentrations; and in spring and summer, with summer showing higher concentrations. For PCT, significant differences were found between spring and summer, as well as winter and summer, with summer again showing higher quantifications.

The instantaneous correlation and cross-correlation of ADD cases and AiV-A viral quantification in RS were analyzed to identify potential temporal relationships between the detection of AiV-A and the occurrence of ADD cases, as this phase reflects the actual circulation of viruses and represents a risk factor associated with disease cases. However, no significant correlation (*p* > 0.05) was identified, as represented in Appendix A.

#### 3.4.2. Sewage Parameters

The sewage parameters, including average flow, total flow, sewage temperature, chemical oxygen demand, and pH at different stages, along with AiV-A quantification and cases of ADD, are described in Table 1.

The Chi-square test indicated significance for pH (*χ*^2^
*=* 49.20; *p* = 0.000), COD (*χ*^2^
*=* 105.96; *p* = 0.000), and AiV-A concentration (*χ*^2^
*=* 53.91; *p* = 0.000). Accordingly, the Kruskal–Wallis test was used, as illustrated in Figure 7.

When comparing the three sewage treatment stages with the correlated variables, the Kruskal–Wallis test with Bonferroni correction showed that a significant difference in pH was observed at all stages, with RS generally exhibiting higher values. Similarly, a statistical difference in COD was found across all stages, with PABT showing the highest values. For AiV-A concentration, a significant difference was also observed when comparing PCT with RS and PABT, with PCT exhibiting the lowest values.

Spearman’s non-parametric correlation was used to assess the relationship between sewage parameters and AiV-A, as shown in Table 2.

In both the RS and PABT phases, a significant correlation with *p* ≤ 0.05 was observed for average flow, with positive correlations (R = 0.320; *p* = 0.021 and R = 0.321; *p* = 0.020, respectively), indicating that higher sewage flow tends to correlate with increased viral concentration. Additionally, in the RS phase, both total flow and ADD cases also showed significant positive correlations (R = 0.310; *p* = 0.025 for total flow and R = 0.342; *p* = 0.013 for ADD cases), suggesting that as these factors increase, so does the concentration. In contrast, a negative correlation was found for COD (R = −0.413; *p* = 0.002), meaning that as COD increases, viral concentration tends to decrease.

In the Mann–Whitney test, used to observe differences between two independent variables, no significant differences (*p* > 0.05) were observed between rainy (*n* = 21) and non-rainy (*n* = 31) days for AiV-A across all three stages of sewage treatment, as shown in Appendix A.

### 3.5. Infectivity Assay of AiV at Different Stages of Sewage Treatment

AiV, isolated in Vero cells (CCL-81), exhibited infectivity across all three stages of sewage treatment, highlighting its ability to replicate and persist even following sewage treatment.

## 4. Discussion

WBE has become a powerful complementary tool for monitoring public health, detecting outbreaks, tracking disease trends, and offering real-time insights, especially in vulnerable communities, reflecting societal development [41]. This study demonstrates that wastewater surveillance can offer a valuable overview of population health, sometimes uncovering unexpected information about pathogens that are not commonly studied locally.

Several studies worldwide have shown the prevalence of AiV in wastewater [17,20,22,42], with some of them also reporting its survival at different stages of sewage treatment [14,21,43,44,45]. However, to the best of our knowledge, this is the first study to report AiV prevalence over a one-year investigation across three different sewage treatment stages, including quantification, molecular characterization, viral infectivity testing, and correlating it with cases of ADD in the studied city and seasonal variations.

AiV was highly detected across all sewage treatment stages, with the lowest detection rate being greater than 70% in the PCT stage throughout the year. This finding surpasses those of Kebe et al. (2021) [14], who reported the presence of the virus in 70% of RS samples and 59.3% in treated samples in Senegal, and corroborates the results of Wang et al. (2020) [21], who reported 88% detection in RS and 63.3% in treated samples in Sweden. Regarding seasonal variation, although the literature suggests no significant seasonality in sewage detections for AiV due to its high detection throughout the year [11], the study by Shaheen and Elmahdy (2024) [46] reported higher detections in winter. The results of this study indicate a significant difference in association with certain seasons, with summer showing higher concentrations of the AiV-A across all three sewage treatment stages analyzed, despite the study by Wang et al. (2020) [21] showing little variability in concentration for the virus. Higher temperatures, an increase in the number of outdoor activities, increased water consumption, higher precipitation, changes in waste and dietary patterns, and increased sewage flow, as observed in this study, are some hypotheses that may lead to an increase in concentration in summer.

Viral concentrations are influenced by factors such as the size of the sewage treatment plant (STP), the population served, pH, and flow rate, among others [47]. In this study, samples were collected from a large STP, characterized by a population served of 480,393 inhabitants, with an average inflow of 108,000 m^3^/day. The mean concentrations of AiV in the positive sewage samples ranged from 2.05 to 4.64 log^10^ GC/mL in RS, 2.31 to 4.72 log^10^ GC/mL in PABT, and 2.13 to 2.85 log^10^ GC/mL in PCT. These findings indicate that, although the virus persisted through the treatment, there was an approximately 2 log^10^ GC/mL reduction in its concentration at the PCT stage compared to RS and the PABT process, in agreement with the review by Farkas et al. (2020) [48]. Furthermore, activated sludge systems have demonstrated effective virus removal, achieving average reductions of up to 3 log^10^ units for non-enveloped viruses, including somatic coliphages, F-specific coliphages, enteroviruses, and noroviruses [49]. A significant difference (*p* ≤ 0.01) was observed, with the lowest AiV-A concentration values recorded in the PCT stage, indicating that, although not complete, some disinfection of the final effluent occurred. Schmitz et al. (2016) [43] suggest that AiV could serve as an indicator of wastewater treatment quality, as it showed a strong correlation with other viral pathogens, albeit at higher concentrations. This suggests that its presence may indicate the potential presence of other pathogenic viruses, as demonstrated in their study. Additionally, according to Farkas et al. (2020) [48], AiV may even be used as a molecular marker for human-derived contamination, as it is widely detected in wastewater and the environment.

Despite typically higher rainfall and increased water consumption for various reasons leading to higher flow rates, which are generally correlated with lower viral concentrations [50], a higher sewage flow was linked to higher concentrations in this study. This result is consistent with the findings of Vallejo et al. (2022) [51], who also observed a positive correlation (R = 0.323) between the average sewage flow and the average daily SARS-CoV-2 concentration. In the study conducted by Bertels et al. (2023) [52], five out of forty STPs also showed higher SARS-CoV-2 concentrations associated with increased flow rates. This suggests that increased organic matter in the sewage system may have carried more viruses, contributing to the higher concentration. Another theory is that sewage overflows and infrastructure failures, worsened by heavy rainfall, can lead to higher levels of waterborne pathogens in watercourses, which has been linked to an increase in gastrointestinal diseases after such periods [53]. On the other hand, the negative correlation suggests a trend of increased COD and a decrease in viral concentration, where COD reflects the amount of organic material present in sewage [54]. It is possible that the viruses are adsorbed to this organic matter [55], which may lead to a lower viral concentration in the system, since pathogen densities are several orders of magnitude higher in sediments compared to the water [56]. Although the secondary treatment of municipal wastewater reduces approximately 47% of the residual COD [56], COD levels were higher in this study at the PABT stage. The removal of COD is also influenced by the temperature of incoming wastewater and the hydraulic retention time [57], which may have contributed to these results. Additionally, it is important to highlight that the collection of these samples was conducted after anaerobic biological treatment, and aerobic biological treatment had not yet been carried out.

In this study, cases of ADD were correlated with AiV-A concentrations in RS to explore a potential relationship, as detectable viruses in wastewater reflect the current circulation of viruses in the area [58,59]. Although ADD cases are not exclusively viral, they were the most appropriate indicator for correlation, given that specific pathogen tests are not routinely conducted. Additionally, globally, ADD remains a significant cause of morbidity and mortality across all age groups, while the clinical role of AiV as gastrointestinal pathogen still remains unclear [11,25]. However, our findings showed no correlation between the number of ADD cases and the detection of AiV-A in RS. One possible explanation for this result is that AiV may contribute to outbreaks through co-infection with other viruses due to its frequent co-detection with other enteric pathogens since the high seroprevalence observed in human populations suggests a clear role for AiV as an enteric agent [11,43,46].

Both AiV genotype A and genotype B are widely distributed globally and may occasionally be found concurrently in the same region, causing gastrointestinal infections [11]. Some findings include AiV-A in Hungary [60], Tunisia [61], Hong Kong [62], and Australia [22]; AiV-B in Venezuela [23], the Netherlands [17], Italy [63], Nepal [64], and South Africa [65]; and both strains (AiV-A and AiV-B) in Finland [66], South Korea [67], and Thailand [68]. In our findings, AiV was molecularly characterized as genotype A in all the samples analyzed. However, when analyzing previous studies, it was observed that the first detection of human AiV in Brazil was reported in stool samples from children suffering from diarrhea [15] in Goiânia, Central Brazil, and was classified as AiV genotype B. Subsequently, in a study published by Portes et al. (2015) [13], which also investigated AiV in children with diarrhea, thirteen samples were identified as genotype A and one as genotype B in Rio de Janeiro, Southeastern Brazil. Interestingly, a regional difference may be occurring; however, further studies conducted simultaneously in different regions and with a larger number of samples should be carried out to confirm this hypothesis.

Known for its established environmental stability, AiV also exhibits a high resistance to low pH levels and is considered one of the most resistant viruses to conventional wastewater treatments, explaining its persistence both in treated and untreated waters [68,69]. From two samples from each stage, the fact that our findings reveal that the virus persists through sewage treatment and still retains its infectious capacity is consistent with the review by Corpuz et al. (2020) [7], which highlights the omnipresence and persistence of viruses in both treated and untreated wastewater. Virus survival in wastewater is strongly affected by environmental factors such as temperature, light, pH, and bacterial and solid contents, along with the type of genetic material (RNA or DNA) and the presence or absence of a viral envelope, which are key factors influencing their persistence [55,70]. In the study conducted by Espinosa et al. (2008) [55], both rotavirus and astrovirus, RNA non-enveloped viruses, showed infectivity and genome persistence in groundwater and surface water, even after chlorination treatment; although a high pH was found to correlate with a reduction in virus titer, this correlation was not observed in this study. On the other hand, the study conducted by Kelly et al. (2022) [71] suggests that low pH may facilitate the exposure of the AiV genome at lower temperatures, as it does not expose its genome at ambient temperature, even at pH 4 [72]. Simmons and Xagoraraki (2011) [73] also describe a 100% cytopathic effect using the BGM cell line in samples from influents and effluents. Although this is not the case in the studied city, it is alarming given that in other places, treated wastewater effluents are used in recreational and agricultural activities, albeit not directly, as well as in the production of potable water from raw sources, potentially posing a risk to public health [7]. In conclusion, this study highlights that much remains to be understood about AiV despite its current characterization in the literature as a potential marker for wastewater pollution. Notably, this is the first study to statistically correlate diarrheal diseases and sewage parameters with AiV, aiming to elucidate the behavior of the virus within a population. While these findings provide valuable insights, they also underscore the need for more comprehensive research to fully explore the dynamics between AiV, environmental factors, and public health. Future studies should focus on expanding these correlations, considering additional variables such as climate, population density, sewage treatment processes, technologies, and operational parameters to offer a more nuanced understanding of the role of AiV in epidemiological surveillance and wastewater management.

## 5. Conclusions

This study marks the first reported detection of AiV-A in sewage samples in Brazil. The prevalence of AiV-A in Brazil underscores the importance of WBE for monitoring pathogens that are not typically studied in certain regions, helping to understand their spread, as well as identifying the pathogens responsible for cases of ADD.

A statistical difference (*p* ≤ 0.05) was observed regarding certain seasons in different sewage stages, with summer showing higher viral concentrations.

Although persistent and infectious after sewage treatment due to its high resistance characteristics, an approximate decrease of 2 log^10^ GC/mL in AiV-A concentration was observed in PCT samples compared to RS and PABT samples. A statistically significant difference was also noted (*p* ≤ 0.01), with lower viral quantities detected in PCT.

Despite the fact that our findings did not demonstrate a statistical correlation (*p* > 0.05) between ADD cases and the detection of AiV-A in RS, further research is warranted to investigate this potential relationship in greater depth, encompassing not only the Aichi virus but also other enteric viruses, in order to better elucidate possible epidemiological correlations.

## Figures and Tables

**Figure 1 viruses-17-00736-f001:**
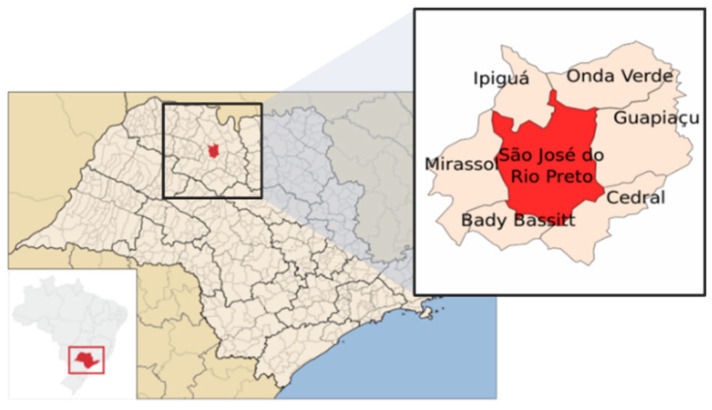
Maps showing the geographical location of the study area, São José do Rio Preto municipality, São Paulo state, Brazil.

**Figure 2 viruses-17-00736-f002:**
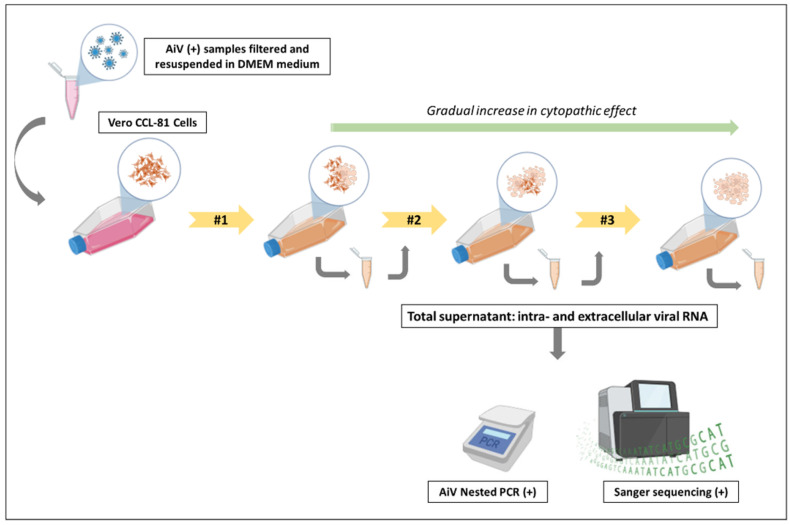
Schematic of the viral infectivity test performed for AiV. Note: Two positive AiV samples from the three different stages of sewage treatment were filtered and resuspended in DMEM medium. They were then inoculated into Vero CCL-81 cells for infection in a Biosafety Level 3 laboratory (BSL-3). After 5 days or upon observation of visible cytopathic effects, the total supernatant, containing intra- and extracellular viral RNA, was collected and re-inoculated into Vero CCL-81 cells, performing the second passage. This cycle was repeated once more to observe cytopathic effects during the third passage. A gradual increase in the cytopathic effect was observed as the passages were performed. After completing the passages, the three supernatants from each treatment stage were removed from the BSL-3 for RNA extraction and testing. The presence of AiV in the RS, PABT, and PCT samples was confirmed through Nested PCR and Sanger sequencing, indicating the ability of the virus to replicate and infect healthy cells.

**Figure 3 viruses-17-00736-f003:**
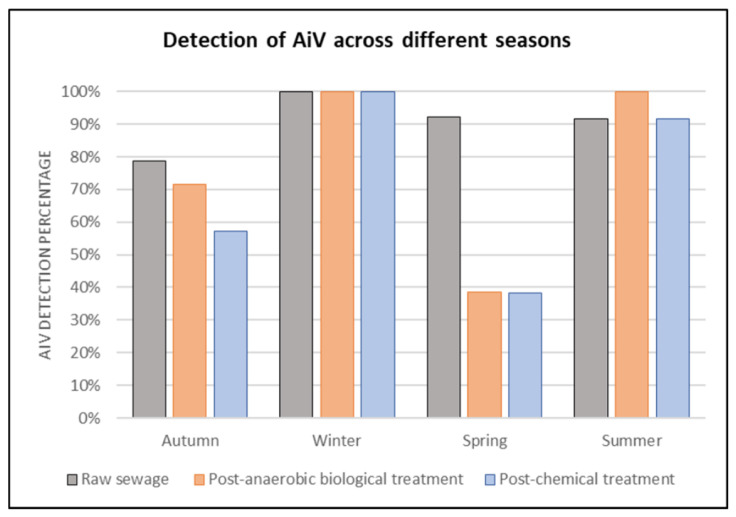
Rate of detection (%) of AiV in RS, PABT, and PCT at the sewage treatment plant in São José do Rio Preto, São Paulo state, grouped by season throughout the year (2022–2023).

**Figure 4 viruses-17-00736-f004:**
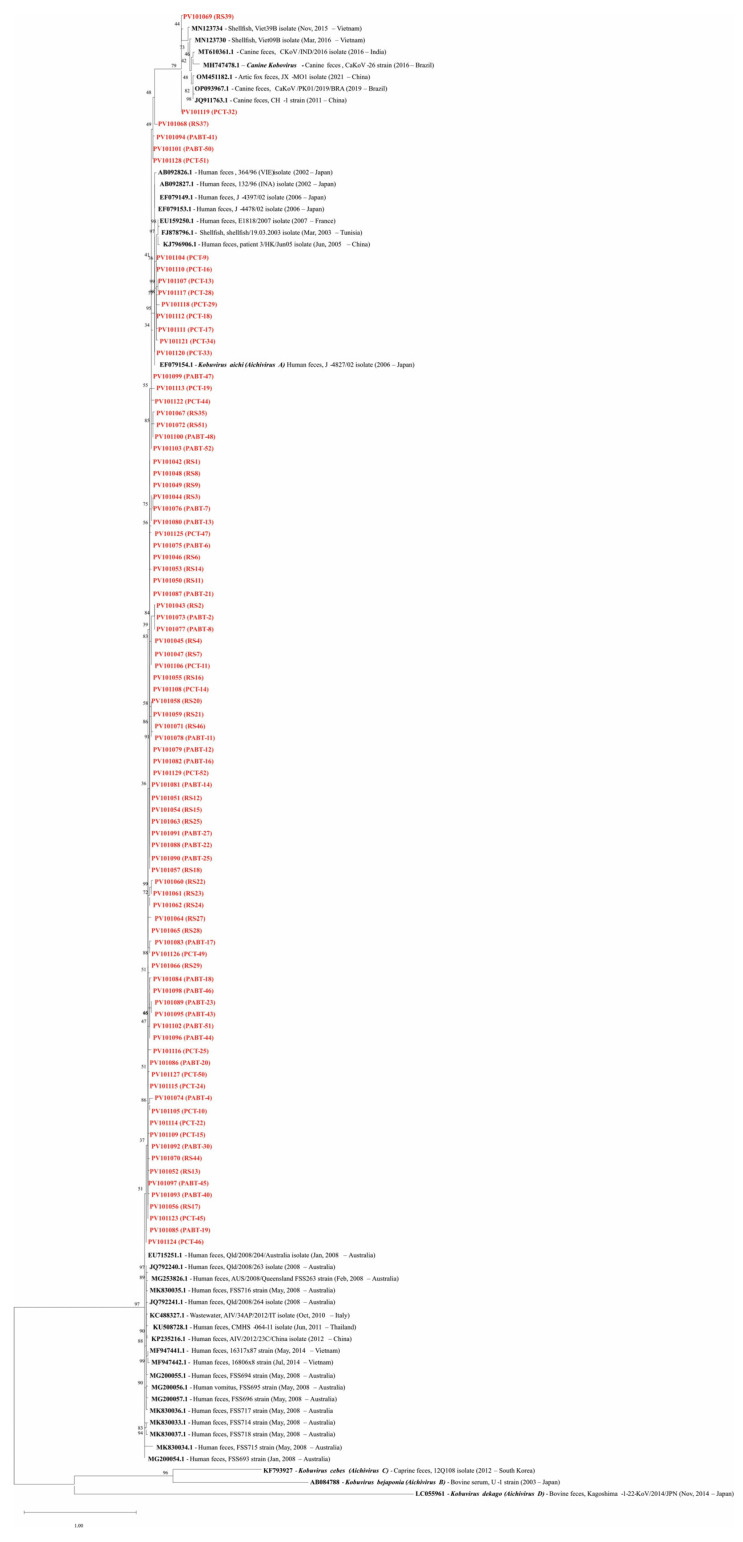
Phylogenetic analysis of AiV performed in the samples based on the 3C region. EF079154.1 is the reference genome for *Kobovirus aichi* (*Aichivirus A*), and the sequences MH747478.1, AB084788, KF793927, and LC055961 were used as external groups, representing the reference genomes of *Canine kobovirus* (AiV-A), *Kobuvirus bejaponia* (*Aichivirus B*), *Kobuvirus cebes* (*Aichivirus C*), and *Kobuvirus dekago* (*Aichivirus D*), respectively. Names in red represent the sequences of this study (Genbank acc. ns. PV101042-PV101129). Bootstrap values (1000 replications) are shown at each branch point. The scale bar represents nucleotide substitutions per site.

**Figure 5 viruses-17-00736-f005:**
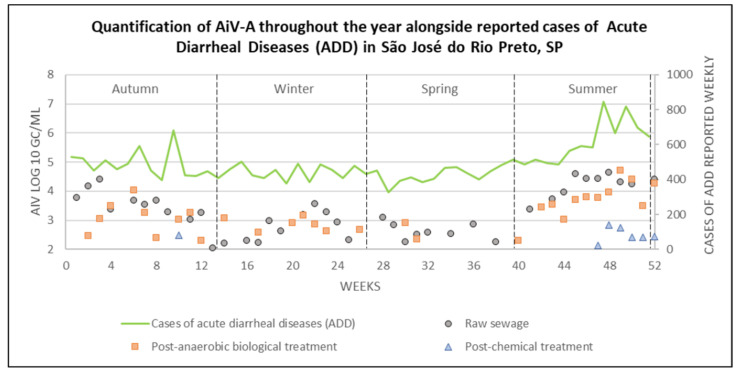
Quantification of AiV-A at each stage of the sewage treatment, alongside reported ADD cases, across different seasons in São José do Rio Preto, São Paulo state (2022–2023).

**Figure 6 viruses-17-00736-f006:**
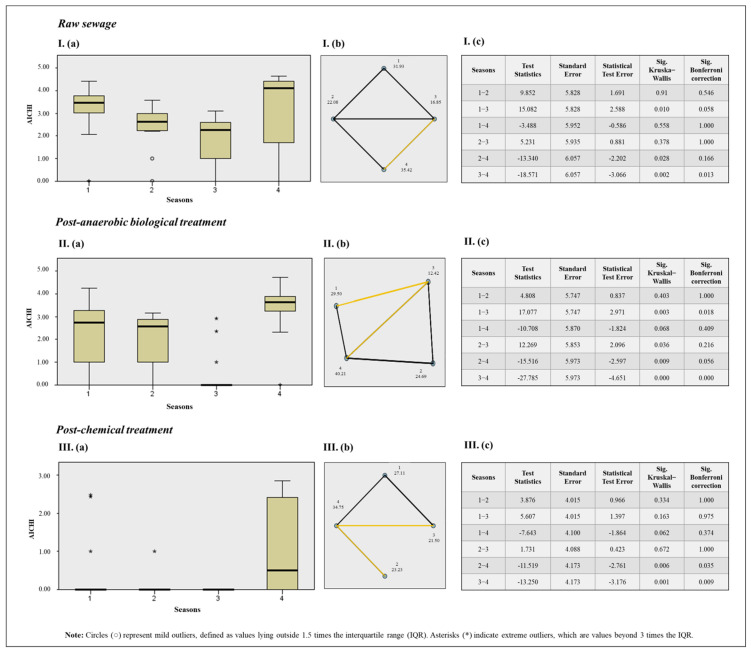
Results of the Kruskal–Wallis test for seasons of the year and AiV-A concentration (**a**). Pairwise comparison between seasons with Bonferroni correction, where statistically significant pairwise differences are indicated by yellow lines, non-significant differences by black lines, and the absence of a line indicates that the comparison was not deemed relevant (**b**). Results table (**c**), where: Season 1: Autumn; Season 2: Winter; Season 3: Spring; Season 4: Summer. I: Results for AiV-A quantification across different seasons in RS samples. II: Results for AiV-A quantification across different seasons in PABT. III: Results for AiV-A quantification across different seasons in PCT samples.

**Figure 7 viruses-17-00736-f007:**
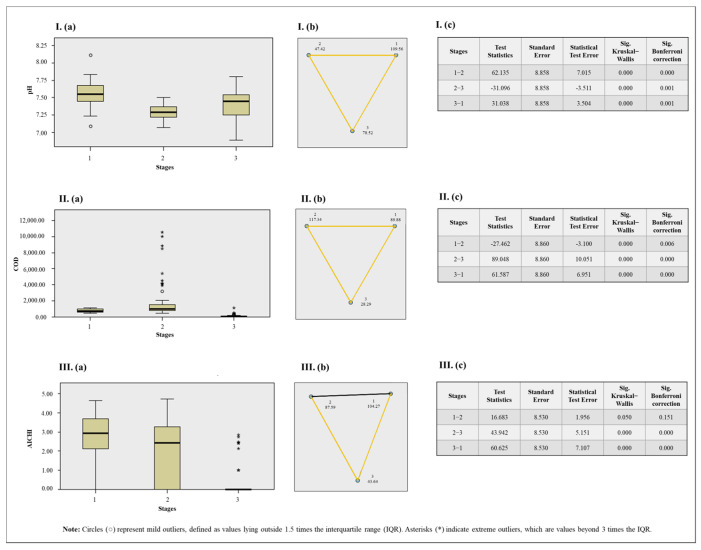
Results of the Kruskal–Wallis test for sewage parameters and stages (**a**). Pairwise comparison between seasons with Bonferroni correction, where statistically significant pairwise differences are indicated by yellow lines and non-significant differences by black line (**b**). Results table (**c**), where Stage 1: Raw sewage; Stage 2: Post-anaerobic biological treatment; Stage 3: Post-chemical treatment. I: Results for pH. II: Results for chemical oxygen demand (COD). III: Results for AiV concentration. Notes: The asymptotic significance values (two-sided test) are presented with a significance level set at 0.05. The *p*-values were adjusted using the Bonferroni correction to account for multiple comparisons.

**Table 1 viruses-17-00736-t001:** Parameters of sewage samples at different stages, AiV-A quantification, and ADD cases.

					Percentiles		
Parameters	Minimum	Maximum	Mean	SD	25th	50th	75th
**Raw sewage**							
AiV (log^10^ GC/mL)	0.00	4.64	2.63	1.42	2.09	2.92	3.69
ADD cases (weekly)	328.00	846.00	491.29	104.76	422.50	464.00	514.50
Average flow (L/s) ^a^	1036.00	1462.00	1182.08	76.13	1133.50	1165.50	1208.00
Total flow (m^3^/day) ^a^	88,515.00	124,147.00	102,063.52	6610.53	97,786.25	101,595.50	105,181.50
Sewage temperature (°C)	22.7	30.5	26.68	1.95	25.10	26.90	28.30
Chemical oxygen demand (mg/L)	456.00	1093.00	757.46	187.13	591.00	738.00	963.50
pH	7.09	8.11	7.56	0.18	7.45	7.55	7.68
**Post-anaerobic biological treatment**							
AiV (log^10^ GC/mL)	0.00	4.72	1.98	1.58	0.00	2.43	3.26
Sewage temperature (°C)	23.10	30.20	26.99	1.83	25.50	27.30	28.30
Chemical oxygen demand (mg/L)	455.0	10,560.00	2037.13	2481.38	808.50	1020.00	1673.50
pH	7.07	7.50	7.29	0.10	7.22	7.29	7.37
**Post-chemical treatment**							
AiV (log^10^ GC/mL)	0.00	2.85	0.39	0.87	0.00	0.00	0.00
Sewage temperature (°C)	21.10	30.20	26.47	2.27	24.40	26.90	28.30
Chemical oxygen demand (mg/L)	18.00	1121.00	128.90	188.19	35.50	50.50	115.25
pH	6.89	7.80	7.41	0.19	7.25	7.45	7.54

SD: Standard deviation; 50th percentile: median; ^a^: Corresponds to the three stages of sewage treatment.

**Table 2 viruses-17-00736-t002:** Spearman’s non-parametric correlation between sewage parameters and AiV-A across the three stages of sewage treatment.

Raw Sewage						
	Average Flow	Total Flow	pH	Temperature	COD	ADD Cases
Correlation Coefficient	0.320 *	0.310 *	0.157	0.127	−0.346 *	0.342 *
Sig. (bilateral)	0.021	0.025	0.267	0.375	0.012	0.013
N	52	52	52	52	52	52
**Post-anaerobic biological treatment**						
	Average flow	Total flow	pH	Temperature	COD	ADD cases
Correlation Coefficient	0.321 *	0.133	0.183	−0.050	−0.413 **	NP
Sig. (bilateral)	0.020	0.349	0.193	0.729	0.002	NP
N	52	52	52	52	52	0
**Post-chemical treatment**						
	Average flow	Total flow	pH	Temperature	COD	ADD cases
Correlation Coefficient	0.121	0.152	−0.271	−0.022	−0.089	NP
Sig. (bilateral)	0.394	0.281	0.052	0.877	0.529	NP
N	52	52	52	52	52	0

*. *p* ≤ 0.05; **. *p* ≤ 0.01; NP: not performed.

## Data Availability

No new data were created.

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
