# Peer review of "Epidemiological and Molecular Surveillance of Aichi Virus A at Different Stages of Sewage Treatment: A One-Year Study in the Southeast of Brazil"

_viruses, 2025, doi:10.3390/v17050736_

Round 1
Reviewer 1 Report
Comments and Suggestions for Authors
The manuscript entitled “Epidemiological and Molecular Surveillance of Aichi virus A at 2 Different Stages of Sewage Treatment: A One-Year Study in the 3 Southeast of Brazil.” This study is very interesting and broadens the knowledge of AiV resistance to depuration treatments. The manuscript is generally well written however there are some comments and mistakes that should be addressed before publication.
Introduction
line 52: put 2 and 5 in superscript in 102 to 105 viral particles
lines 58-60: It would be appropriate to specify where this is valid: in the world? in the USA in Europe? you cited a French work and this is a bit limiting, I suggest you refer to a review or more studies in more countries in the world.
lines 90-94: These are results that in my opinion should not be anticipated at the end of the discussion, this is not the abstract. I would remove this sentence.
Materials and methods
Lines 117-118: The explanation of sampling is a bit confusing I suggest you explain it better. In line 109 you say that twenty four 500ml samples are taken every hour in three points, then in line 112 that sampling is done weekly for one year and then in lines 115-116 that you take a sample every 3000m3 resulting in 36 events x days. It seems to me that there is excessive information in the end the information that is needed is that "sampling was conducted weekly for one year (52 weeks)" and "a total of 156 samples were collected".
Results
Line 249: missing at where you obtained 32.8% of positive samples….. 41 (32.8%) from, and
Line 251:…. the detection rates were 90.4%, 78.8%, and 71.1% for the same sewage treatment 251 stages, respectively……. It is not clear what these percentages refer to. Please explain better.
Line 304:…. Out of the 125 samples positive by Nested PCR, 91 were positive by qPCR…… Since real time PCR is more sensitive than nested PCR, how do you explain why you found fewer positives with real time? Maybe because nested amplifies more genotypes while Real Time is specific for genotype A? If so, specify it in the materials and methods.
Lines 320-321: ….. Descriptive analyses of the association between seasons and AiV-A quantification are 320 reported in Supplementary Table ST4. For non-parametric tests, percentile values may be 321 used to assess the data distribution or rankings…… in table ST4 there is no statistical association, perhaps you should change the term association because it can be misinterpreted.
Figure 7: This figure is not a result but an explanation of the methods used, in my opinion you should move it to M&M or remove it.
Supplementary files
in caption of Figure 2 you should correct with Figure SF2.
line 11: the caption of Table ST1 is attached to that of figure 2.
Author Response
Comments 1: Line 52: put 2 and 5 in superscript in 102 to 105 viral particles.
A: Thank you for your observation. The correction has been made and is still reflected in line 52.
Comments 2: Lines 58-60: It would be appropriate to specify where this is valid: in the world? in the USA in Europe? you cited a French work and this is a bit limiting, I suggest you refer to a review or more studies in more countries in the world.
A: Thank you for the suggestion. Yes, these pathogens are frequently cited as the main causes of gastroenteritis worldwide, and this is well established in the literature. However, as recommended, we reviewed additional papers to further support and strengthen this statement. You can still see the changes in lines 58-60.
Comments 3: Lines 90-94: These are results that in my opinion should not be anticipated at the end of the discussion, this is not the abstract. I would remove this sentence.
A: Thank you for your comment. We accept the suggestion and remove the sentence:
“To the best of our knowledge, this is the first study to examine AiV in sewage samples in Brazil and represents the most comprehensive investigation of AiV in such samples to date. The findings highlight the persistence of AiV within sewage treatment systems, emphasizing the need for continued monitoring and further research into its potential implications for public health.”
Comments 4: Lines 117-118: The explanation of sampling is a bit confusing I suggest you explain it better. In line 109 you say that twenty four 500ml samples are taken every hour in three points, then in line 112 that sampling is done weekly for one year and then in lines 115-116 that you take a sample every 3000m3 resulting in 36 events x days. It seems to me that there is excessive information in the end the information that is needed is that "sampling was conducted weekly for one year (52 weeks)" and "a total of 156 samples were collected".
A: Thank you for pointing that out. We accepted the suggestion. We removed the part regarding the sampler:
“It collects one sample for every 3000 m³ processed by the plant, resulting in approximately 36 sampling events per day.” We agree that it was too detailed and not essential to the study.
It is important to emphasize that the sampling was carried out over a total period of 24 hours though — that is, small volumes were collected throughout the day rather than in a single grab sample. This detail is relevant, especially when evaluating pathogen concentration and, ultimately, its detection frequency, which was weekly. For this reason, we decided to keep this information. You can now find this revised section in lines 106-112.
Comments 5: Line 249: missing at where you obtained 32.8% of positive samples….. 41 (32.8%) from, and
A: Thank you for the comment. We have corrected it, and the changes can be seen in line 259.
Comments 6: Line 251: …. the detection rates were 90.4%, 78.8%, and 71.1% for the same sewage treatment stages, respectively……. It is not clear what these percentages refer to. Please explain better.
A: Thank you for your question. While lines 258-260 present all positive samples along with the percentage corresponding to each treatment stage, lines 260-262, which you refer to in your comment, consider each treatment stage separately. In other words, we calculated how many of the 52 samples analyzed at each stage tested positive, rather than grouping them together.
Comments 7: Line 304:…. Out of the 125 samples positive by Nested PCR, 91 were positive by qPCR…… Since real time PCR is more sensitive than nested PCR, how do you explain why you found fewer positives with real time? Maybe because nested amplifies more genotypes while Real Time is specific for genotype A? If so, specify it in the materials and methods.
A: Thank you for your question. In this study, we found that Nested PCR was able to detect lower viral quantification than qPCR, which is expected due to the two amplification cycles involved in Nested PCR and is likely not attributable to genotyping differences — especially considering that the post-chemical treatment samples were the most affected.
This interpretation is further supported when analyzing the samples as a whole. As shown in Supplementary Table ST3, among the samples positive by Nested PCR but negative by qPCR, only 2 were from raw sewage, 5 from post-anaerobic biological treatment, and 27 from post-chemical treatment. This suggests that viral concentration decreased throughout the treatment process, becoming undetectable by qPCR. Additionally, although Nested PCR can amplify genotypes beyond A, sequencing of the amplified products identified only genotype A. Quantification by qPCR was just performed after this sequencing confirmation.
Comments 8: Lines 320-321: ….. Descriptive analyses of the association between seasons and AiV-A quantification are reported in Supplementary Table ST4. For non-parametric tests, percentile values may be used to assess the data distribution or rankings…… in table ST4 there is no statistical association, perhaps you should change the term association because it can be misinterpreted.
A: Thank you for your suggestion. We have revised the sentence to:
“Descriptive analyses of AiV-A quantification across seasons are reported in Supplementary Table ST4.” This change has been implemented in line 339.
Comments 9: Figure 7: This figure is not a result but an explanation of the methods used, in my opinion you should move it to M&M or remove it.
A: Thank you for pointing that out. We agreed with your suggestion and have moved the figure to the Materials and Methods section. It is now labelled as Figure 2 and can be found at line 195.
Supplementary files
Comments 10: in caption of Figure 2 you should correct with Figure SF2.
A: Thank you for your observation. The caption has been corrected and can be seen in line 8 of the Supplementary Material.
Comments 11: line 11: the caption of Table ST1 is attached to that of figure 2.
A: Thank you for your comment. This has been corrected, as reflected between the lines 10-11 of the Supplementary Material.
Reviewer 2 Report
Comments and Suggestions for Authors
The work has been performed at a high level, and all methods are appropriate for achieving the stated aims. The topic is highly relevant not only for analyzing local data but also for the global virology community. I would like to commend the authors for their meticulousness—the text is clear and well-structured. Additionally, the statistical analysis of the data enhances confidence in the obtained results. The authors have done extensive work, conducting various types of analyses (Nested PCR, qPCR, chemical analysis of sewage, viral infectivity testing). The fact that samples were collected continuously throughout the year is particularly important.
Line 27 – "The prevalence of AiV in raw sewage samples was…"
I think it should specify "throughout the year." It wasn’t immediately clear that this refers to the entire study period rather than peak values.
Line 52 – "... 102 to 105 viral…" – Format the exponents correctly.
Line 165 – You mention that "...3C region sequences of AiV were selected from GenBank…"
But what was the similarity with the closest relatives?
Line 167 – I assume that model selection using ModelFinder was performed but simply not described in the text, correct? Were other alignment algorithms used, such as those implemented in MAFFT? And was the alignment checked?
Line 195 – Did you examine the supernatant by TEM?
Line 203 – "...from 1 × 106 to 1 × 101 genomic…" – It’s better to list from smallest to largest (i.e., 1 × 10¹ to 1 × 10⁶). The same applies to Line 31 (smallest to largest).
Line 249 – "...41 (32.8%) from, …" – "PABT" is likely missing here.
Figure 3 – The resolution of the figure is low, and I recommend increasing the font size for bootstrap supports.
Line 270 – "The scale bar represents nucleotide substitutions per site."
This sentence should probably be moved to the figure legend.
Supplementary
Line 11 – "Supplementary Table ST1" – Move to a new line.
Line 16 – "Table ST2" – The formatting is inconsistent. Above, it’s written as "Supplementary Table ST1." Please make it uniform.
Author Response
Comments 1: Line 27 – "The prevalence of AiV in raw sewage samples was…"
I think it should specify "throughout the year." It wasn’t immediately clear that this refers to the entire study period rather than peak values.
A: Thank you for your comment. We have added "throughout the year" as suggested and you can see it in lines 27-28.
Comments 2: Line 52 – "... 102 to 105 viral…" – Format the exponents correctly.
A: Thank you for your observation. The correction has been made and is still reflected in line 52.
Comments 3: Line 165 – You mention that "...3C region sequences of AiV were selected from GenBank…" But what was the similarity with the closest relatives?
A: Thank you for your question. We have included in the manuscript the range of sequence similarity between the study samples and the corresponding reference sequences. For Kobuvirus Aichi (EF079154.1) and the human samples, the similarity ranged from 88.8% to 97.6%. For Canine Kobuvirus (MH747478.1) and the canine samples, the sequence similarities were 90.1% and 91.1%. The lower similarity observed in some samples is likely due to the presence of degenerate bases.
Additionally, we have provided a table comparing these sequences with other reference sequences for further consultation. This addition can be found in lines 294–302 of the revised manuscript.
Comments 4: Line 167 – I assume that model selection using ModelFinder was performed but simply not described in the text, correct? Were other alignment algorithms used, such as those implemented in MAFFT? And was the alignment checked?
A: Thank you for your question. Yes, the phylogenetic analysis was performed using the IQ-TREE, which includes ModelFinder to automatically select the best-fit substitution model during the Maximum Likelihood inference. We have incorporated this information into the text for greater clarity (line 165). As for the alignment, MUSCLE was used, as previously described in lines 162–163. No other alignment method, such as MAFFT, was used.
Comments 5: Line 195 – Did you examine the supernatant by TEM?
A: Thank you for your question. No, we did not examine the supernatant by TEM, as this methodology is very expensive, labor-intensive, and requires specific preparation. In the infectivity test, the goal was to assess whether the virus remained resistant to treatment while retaining its replicative and infectious cycle. This was observed through the cytopathic effect on the cells and confirmed in each passage by Nested PCR, which showed robust amplification. These results indicate that the detected viral RNA did not originate from residual RNA of the original sample, but rather from actively replicating, infectious viral particles.
Comments 6: Line 203 – "...from 1 × 106 to 1 × 101 genomic…" – It’s better to list from smallest to largest (i.e., 1 × 10¹ to 1 × 10⁶). The same applies to Line 31 (smallest to largest).
A: Thank you for your comment. We have accepted your suggestion, and you can see the changes in lines 31-32, and 213.
Comments 7: Line 249 – "...41 (32.8%) from, …" – "PABT" is likely missing here.
A: Thank you for pointing this out. We have added “PABT” and you can see it now in the line 259.
Comments 8: Figure 3 – The resolution of the figure is low, and I recommend increasing the font size for bootstrap supports.
A: Thank you for your observation. We believe that the loss of image quality may be due to the large file size during automatic PDF generation, or possibly because the image was zoomed in without properly enlarging it within the document. However, the original file is of excellent resolution (600 dpi), and once published, it will allow for clearer and more detailed viewing.
Comments 9: Line 270 – "The scale bar represents nucleotide substitutions per site."
This sentence should probably be moved to the figure legend.
A: Thank you for your suggestion. We have moved the sentence to the figure legend and now you can see it in lines 292-293.
Supplementary
Comments 10: Line 11 – "Supplementary Table ST1" – Move to a new line.
A: Thank you for your comment. This has been corrected, as reflected the line 11 of the Supplementary Material.
Comments 11: Line 16 – "Table ST2" – The formatting is inconsistent. Above, it’s written as "Supplementary Table ST1." Please make it uniform.
A: Thank you for pointing this out. We have reviewed and corrected all caption formatting, and you can now find them in lines 2, 8, 11, 15, 19, 27, and 30 of the Supplementary Material.
Reviewer 3 Report
Comments and Suggestions for Authors
The manuscript by Mariah Cristina Antunes do Nascimento and colleagues presents a study on the prevalence of Aichi virus (AiV) at different stages of sewage treatment in São José do Rio Preto. The authors conducted a comprehensive analysis using PCR, sequencing, and bioinformatic methods. The manuscript is logically structured, follows a conventional format, and the illustrations are appropriate. The detailed methodology contributes to the robustness of the study. The findings are potentially of interest to readers and may be valuable for epidemiological studies of AiV. With some revisions, the manuscript is suitable for publication in Viruses.
Specific comments:
Line 24: “The data were also correlated” - I suggest revising this statement. Given that no statistically significant correlation was found, a more accurate phrasing would be: “We explored the potential for correlation…” or “Correlation analysis was performed…”
Line 31: “ranged from 4.64 to 2.05 log10 GC/mL” - For consistency, I recommend presenting the range in ascending order: “from 2.05 to 4.64 log10 GC/mL.” This applies to other instances throughout the abstract, results, and discussion sections.
Line 52: “discharging 102 to 105 viral particles” - There appears to be a formatting error here. Please correct.
Line 249: Please specify the method used to determine that 41 (32.8%) of the tested samples were positive.
Lines 269-271: Please relocate the description of BS and related information to the figure caption for Figure 2.
Figure 3: The labels are difficult to read. Please improve the figure quality and ensure labels are legible.
Figures 5 and 6: I recommend providing additional descriptive information for the (b) graphs in the figure captions.
Author Response
Comments 1: Line 24: “The data were also correlated” - I suggest revising this statement. Given that no statistically significant correlation was found, a more accurate phrasing would be: “We explored the potential for correlation…” or “Correlation analysis was performed…”
A: Thank you for pointing that out. We have accepted your suggestion and revised the sentence to:
“The data were also analyzed for potential correlations.” Please see lines 24–25.
Comments 2: Line 31: “ranged from 4.64 to 2.05 log10 GC/mL” - For consistency, I recommend presenting the range in ascending order: “from 2.05 to 4.64 log10 GC/mL.” This applies to other instances throughout the abstract, results, and discussion sections.
A: Thank you for your comment. We have accepted your suggestion, and you can see the changes in lines 31-32, 213, 325, 329, 333, 448-449.
Comments 3: Line 52: “discharging 102 to 105 viral particles” - There appears to be a formatting error here. Please correct.
A: Thank you for your observation. The correction has been made and is still reflected in line 52.
Comments 4: Line 249: Please specify the method used to determine that 41 (32.8%) of the tested samples were positive.
A: Thank you for your comment. We believe you intended to suggest adding the specific treatment stage related to the 41 (32.8%) result, and we have now included it (PABT). The method used to determine this positivity was Nested PCR, which is also mentioned in the same line. Please see line 259.
Comments 5: Lines 269-271: Please relocate the description of BS and related information to the figure caption for Figure 2.
A: Thank you for your suggestion. We have moved the sentence to the figure legend and now you can it in line 292-293.
Comments 6: Figure 3: The labels are difficult to read. Please improve the figure quality and ensure labels are legible.
A: Thank you for your observation. We believe that the loss of image quality may be due to the large file size during automatic PDF generation, or possibly because the image was zoomed in without properly enlarging it within the document. However, the original file is of excellent resolution (600 dpi), and once published, it will allow for clearer and more detailed viewing.
Comments 7: Figures 5 and 6: I recommend providing additional descriptive information for the (b) graphs in the figure captions.
A: Thank you for your suggestion. We agree and have added further information, as can be seen in lines 349-351 and 380-382.
Round 2
Reviewer 1 Report
Comments and Suggestions for Authors
The manuscript titled "Epidemiological and Molecular Surveillance of Aichi virus A at Different Stages of Sewage Treatment: A One-Year Study in the Southeast of Brazil" has been revised and the authors have modified it based on the suggestions they have been given. Here are the last small suggestions to further improve it before publication:
Discussion
Line 416: you already explained Wastewater-based epidemiology in the introduction you can put the abbreviation WBE here.
Line 427: you already explained acute diarrheal diseases in the introduction you can put the abbreviation ADD here.
Conclusion
Lines 558-559: the phrase “a more nuanced understanding of AiV's role in epidemiological surveillance and wastewater management” is a repetition already present at the end of the discussion, I suggest you to change it.
Author Response
Discussion
Comments 1: Line 416: you already explained Wastewater-based epidemiology in the introduction you can put the abbreviation WBE here.
A: Thank you for your observation. The correction has been made and is still reflected in line 416.
Line 427: you already explained acute diarrheal diseases in the introduction you can put the abbreviation ADD here.
A: Thank you for your observation. The amendment has been implemented and continues to be reflected at line 427.
Conclusion
Lines 558-559: the phrase “a more nuanced understanding of AiV's role in epidemiological surveillance and wastewater management” is a repetition already present at the end of the discussion, I suggest you to change it.
A: Thank you for pointing that out. We accepted the suggestion. The modifications can be found in lines 556-559.